# Linear Skin Defects with Multiple Congenital Anomalies (LSDMCA): An Unconventional Mitochondrial Disorder

**DOI:** 10.3390/genes12020263

**Published:** 2021-02-11

**Authors:** Alessia Indrieri, Brunella Franco

**Affiliations:** 1Telethon Institute of Genetics and Medicine (TIGEM), Via Campi Flegrei, 34, 80078 Pozzuoli, Naples, Italy; indrieri@tigem.it; 2Institute for Genetic and Biomedical Research (IRGB), National Research Council (CNR), 20090 Milan, Italy; 3Medical Genetics, Department of Translational Medical Sciences, University of Naples “Federico II”, Via Sergio Pansini 5, 80131 Naples, Italy

**Keywords:** MLS/MIDAS/LSDMCA, X-inactivation, HCCS, COX7B, NDUFB11, mitochondrial disorders, mitochondrial respiratory chain, microphthalmia, linear skin defects

## Abstract

Mitochondrial disorders, although heterogeneous, are traditionally described as conditions characterized by encephalomyopathy, hypotonia, and progressive postnatal organ failure. Here, we provide a systematic review of Linear Skin Defects with Multiple Congenital Anomalies (LSDMCA), a rare, unconventional mitochondrial disorder which presents as a developmental disease; its main clinical features include microphthalmia with different degrees of severity, linear skin lesions, and central nervous system malformations. The molecular basis of this disorder has been elusive for several years. Mutations were eventually identified in three X-linked genes, i.e., *HCCS*, *COX7B*, and *NDUFB11*, which are all endowed with defined roles in the mitochondrial respiratory chain. A peculiar feature of this condition is its inheritance pattern: X-linked dominant male-lethal. Only female or XX male individuals can be observed, implying that nullisomy for these genes is incompatible with normal embryonic development in mammals. All three genes undergo X-inactivation that, according to our hypothesis, may contribute to the extreme variable expressivity observed in this condition. We propose that mitochondrial dysfunction should be considered as an underlying cause in developmental disorders. Moreover, LSDMCA should be taken into consideration by clinicians when dealing with patients with microphthalmia with or without associated skin phenotypes.

## 1. Introduction: An Historical Perspective 

In the early 1990s, females with a specific phenotypic combination of short stature with developmental abnormalities, including microphthalmia and linear skin defects of the face and neck, were reported. In these cases, abnormalities of the short arm of the X-chromosome resulted in monosomy of the Xp22.3 region [1]. In males, the lack of distal Xp results in contiguous gene syndromes characterized by the appearance of recessive traits underlain by a number of genes disrupted by the X-chromosome abnormalities. However, apart from eight males showing a 46 XX karyotype and a translocation Xp/Yp, no males with the typical ocular and skin phenotype have been described to date. The condition defined by the clinical features observed in females was denominated Microphthalmia with Linear Skin defects (MLS) syndrome (MIM # 309801). Given the presence of aborted fetuses in familial cases and the absence of males with X-chromosomal rearrangements and a classical MLS phenotype, it was defined as an X-linked dominant male-lethal trait (alternatively named Microphthalmia Dermal Aplasia and Sclerocornea (MIDAS)). This condition was subsequently demonstrated to be genetically heterogeneous and, following identification of three responsible genes, has been renamed Linear Skin Defects with Multiple Congenital Anomalies (LSDMCA) 1, 2, and 3 (OMIM # 309801, OMIM # 300887, and OMIM # 300952), respectively. 

Deletion mapping combined with a positional candidate gene strategy allowed the identification of the first disease gene, *HCCS* (holocytochrome c synthase; OMIM # 300056), from the Xp22.2 region in 2006 [2]. Subsequently, in 2012, a candidate gene strategy led to the identification of mutations in *COX7B* (cytochrome c oxidase subunit 7B; OMIM # 300805) localized to Xq21.1 [3]. Finally, in 2015, whole-exome sequencing facilitated the detection of point mutations in *NDUFB11* (NADH–ubiquinone oxidoreductase 1 beta, subcomplex 11; OMIM # 300403) from the Xp11.3 region [4], thus placing all the causative genes so far identified for this disease on the X chromosome. The disorder is quite rare; to date, a total of 82 affected individuals, plus four aborted and molecularly diagnosed fetuses, have been described. The majority of cases account for LSDMCA1 and are associated with abnormalities of distal Xp (70 cases with Xp22 translocation/terminal deletion/small or large interstitial deletions). In addition, point mutations in the *HCCS* gene have also been described. The remaining cases consist of point mutations in *COX7B* and *NDUFB11* (see Table 1). 

Notably, there is still room for additional causative genes as mutation analysis has demonstrated that not all reported cases can be explained by the causative genes so far described (BF, unpublished).

## 2. The Molecular Basis of LSDMCA

As mentioned, the majority of LSDMCA1 patients carry rearrangements of the Xp region and are females. Exceptions to this observation are represented by eight male individuals showing 46 XX karyotypes and translocations involving the short arms of the X and Y chromosomes and resulting in Xp monosomy in one of two X chromosomes [5]. A complete list of chromosomal abnormalities associated with LSDMCA1 is reported in Appendix A. The LSDMCA1 critical region was defined by characterization of patients with deletions and translocations involving the short arm of the X chromosome [6,7] and spans approximately 610 Kb in Xp22.2. The region contains three genes: *MID1*, which has been shown to be responsible for X-linked Opitz syndrome [8]; *HCCS*, which encodes for the mitochondrial holocytochrome c-type synthase (known as heme lyase) that catalyzes the covalent attachment of heme to both apo-Cytochrome (Cyt) c and c1 [9,10], and *ARHGAP6* [11]. In 2002, it was shown that deletions involving the syntenic LSDMCA1 critical region in the mouse led to embryonic lethality, which could be rescued by overexpression of the human holocytochrome c-type synthase, indicating *HCCS* as the most convincing candidate gene for LSDMCA1 [12]. 

However, the most conclusive evidence was derived from analysis of the few LSDMCA cases not associated with chromosomal abnormalities [2,3,4]) (Table 1 and Appendix A). First, point mutations and a small deletion were identified in *HCCS*, thus providing the evidence that this gene is indeed responsible for LSDMCA1 [2]. Specifically, de novo heterozygous point mutations, i.e., a nonsense mutation (c.589C > T/p.R197*) which was subsequently identified in an additional case [13] and a missense mutation (c.649C > T/p.R217C), were identified in two patients showing a normal karyotype [2]. Later, a novel missense mutation (c.475G > A/p.E159K) was identified in a sporadic case with bilateral microphthalmia and sclerocornea without skin lesions, indicating that the phenotypic variability described in LSDMCA1 is not correlated to the extent of the Xp deletion [14]. Finally, a mosaic 2-bp *HCCS* deletion, (c.[=/524_525delAG] (p.[=/E175Vfs*30]), was identified in a patient with unilateral ocular anomalies and no skin defects [13]. This patient showed a variable degree of mosaicism that may have contributed to her mild phenotype. However, a patient with a mosaic X-chromosomal rearrangement showed the classical LSDMCA1 phenotype [15], indicating that other mechanisms are responsible for the high clinical variability in patients with *HCCS* mutations.

HCCS is a highly conserved, nuclear-encoded mitochondrial protein, located on the outer surface of the inner mitochondrial membrane, where it catalyzes the covalent attachment of heme to both Cytochrome (Cyt) c and c1 [9,10]. Cytc1 is an integral component of the mitochondrial respiratory chain (MRC) complex III and transfers electrons to Cytc, which, in turn, shuttles them from complex III to IV. In *Saccharomyces cerevisiae*, two heme lyases exist, Cyc3 and Cyt2, responsible for heme incorporation into Cytc and Cytc1, respectively [16]. Inactivation of either Cyc3 or Cyt2 results in loss of respiratory growth [17,18]. Conversely, in higher eukaryotes, a single heme lyase, HCCS, is sufficient for Cytc and Cytc1 maturation [9,16]. Functional studies have demonstrated that the point mutations interfere with the role of HCCS in mitochondrial function and exert their pathogenic effect via oxidative phosphorylation (OXPHOS) impairment [19]. Moreover, conditional inactivation of *Hccs* in the murine heart and its downregulation in medaka fish result in severe OXPHOS defects, thus definitively demonstrating a key role for this protein in MRC formation and function [19,20]. 

Although HCCS is an ubiquitous protein, the phenotype observed in LSDMCA1 is restricted to specific organs, suggesting that its dosage/function may be critical in selected tissues. A critical role for HCCS in the control of apoptosis activation has been described. In the intrinsic programmed cell death pathway, mitochondrial outer membrane permeabilization leads to the release of Cytc that binds cytosolic Apaf1 and recruits and activates the initiator caspase-9 to form the apoptosome complex [21,22]. Moreover, other proapoptotic proteins, such as SMAC/DIABLO and Omi/HtrA2, are released in the cytosol and bind inhibitors of apoptosis (IAPs), relieving their inhibitory effects on caspase activity [23]. In injured adult rat motor neurons, Hccs migrates from mitochondria to the cytosol under apoptotic stimuli, resulting in suppression of the X-linked inhibitor of apoptosis (XIAP) protein and activation of caspase-3, indicating that the cytosolic HCCS may act, similarly to SMAC/DIABLO and Omi/HtrA2, as an IAP binding protein [24]. Interestingly, downregulation of *hccs* in medaka fish leads to increased cell death via apoptosome-independent caspase-9 activation, which occurs in the mitochondria and is triggered by OXPHOS defects and overproduction of reactive oxygen species (ROS). Notably, the activation of this pathway specifically occurs in the brain and eyes and underlies the development of microphthalmia and microcephaly observed in LSDMCA1 [19]. On the other hand, increased cell death was not detected in the hearts of HCCS deficient models (mice and fish) whereas decreased cardiomyocyte proliferation was observed in *Hccs*^+/−^ mutant hearts [19,20].

Almost all LSDMCA cases (without chromosomal abnormalities) show skewed X-chromosome inactivation, which suggests that there is a selective disadvantage for cells carrying the mutated allele on their active X [25]. Moreover, the increased cell death observed in the central nervous system (CNS) [19] and the reduced proliferation of *Hccs*-deficient cardiomyocytes [20] suggest that the tissue-specific activation of different molecular pathways may cause some of the phenotypes observed in LSDMCA patients, explaining the specificity of the defects observed. On the other hand, the influence of X-inactivation remains the best explanation for the high degree of clinical variability observed in LSDMCA patients.

After the discovery of *HCCS* mutations, more recent data have also implicated the X-linked *COX7B* and *NDUFB11* genes in the pathogenesis of this genetic disorder. Interestingly, these genes are key components of the MRC complexes IV and I, respectively.

*COX7B* is a small gene comprising three exons on Xq21.1. It is ubiquitously expressed and encodes an integral component of the cytochrome c oxidase (COX), the MRC complex IV [26,27]. In humans, COX is composed of three proteins encoded by the mitochondrial DNA (mtDNA) (COX1, COX2, and COX3) that assemble with 10 nuclear-encoded proteins (COX4, COX5A, COX5B, COX6A, COX6B, COX6C, COX7A, COX7B, COX7C, and COX8) to form the mature holo-complex [28]. 

In 2012, pathogenic point mutations in *COX7B* were found in LSDMCA patients with normal karyotypes and no mutations in *HCCS* [3]. In particular, a heterozygous 1-bp deletion in exon 3 (c.196delC/p.L66Cfs*48), a heterozygous splice mutation in intron 1 (c.41-2A > G/p.V14Gfs*19), and a heterozygous nonsense mutation in exon 2 (c.55C > T/p.Gln19*) were identified [3] (see Table 1). These mutations result in a truncated protein, which is predicted to lack the functional domain necessary for interaction with other subunits of the COX complex [3]. Although the MRC complex IV has been extensively studied, the function of COX7B within this complex has only been characterized after the discovery of the mutations leading to LSDMCA2. Notably, it has been shown that the small COX7B subunit is necessary for COX activity, COX assembly, and mitochondrial respiration [3]. Moreover, downregulation of *cox7B* in medaka fish resulted in increased cell death, leading to microcephaly and microphthalmia, thus resembling the phenotype observed in *hccs*-defective fish [3,29]. These data indicate an essential function for complex IV activity in vertebrate CNS development [3].

*NDUFB11* is located on Xp11.23 and comprises three exons. Moreover, this gene is ubiquitously expressed and encodes for one of the 30 supernumerary subunits of NADH:ubiquinone oxidoreductase, the MRC complex I [30,31]. This complex is the largest within the MRC and is composed of around 45 subunits in mammals, seven of which are encoded by mtDNA. Only 14 proteins represent the core subunits and are essential for energy transduction [31,32]. 

In 2015, a heterozygous nonsense mutation (c.262C > T/p.Arg88*) and a heterozygous 1-bp deletion leading to frameshift (c.402delG/p.Arg134Serfs*3) in *NDUFB11* were described in LSDMCA patients [4] (see Table 1). The authors also showed that NDUFB11 is necessary for the assembly of complex I membrane arm, for the maturation of the holocomplex, and for complex I-dependent mitochondrial respiration [4]. Interestingly, *NDUFB11* knockdown in HeLA cells caused impaired cell growth and increased apoptosis [4], also shown in in vivo models of *HCCS* and *COX7B* downregulation [19,29].

## 3. The Clinical Spectrum of LSDMCA

The characterization of patients has facilitated a thorough description of the clinical spectrum observed in this condition. Appendix A depicts all the LSDMCA cases reported to date, with descriptions of the commonly observed clinical findings, which are summarized in Table 2. 

Linear skin lesions. The most constant and archetypal clinical feature is represented by the linear skin lesions. These are present in the majority of cases, regardless of the underlying causative mutation (77% of LSDMCA1 with Xp22 rearrangements, 40% of cases with *HCCS* point mutations, 100% of LSDMCA2 and 67% of LSDMCA3 patients) (Table 2). They are commonly seen at birth as irregular linear erythematous patches, sometimes covered by hemorrhagic crusts. The skin marks are usually located on the face, particularly the cheeks, and the neck, with an asymmetric distribution and often extending to the chin and the nose (Figure 1). 

The same lesions can rarely be seen on the hands and other parts of the body. The cutaneous signs follow Blaschko lines and tend to improve over time, leaving minimal to no residual scars. Histopathological investigation revealed a thin, atrophic epidermis lacking rete ridges with a significant infiltrate of lymphocytes. In addition, irregular bundles of smooth muscle were observed in the deep dermis while adnexal structures were missing [49]. In a different study, dermatoscopic examination demonstrated erythematous areas with telangiectasias accompanied by an absence of sebaceous glands and vellus hairs, thus confirming the histopathological findings [64]. Notably, all investigations performed on the cutaneous wounds were done on LSDMCA1 patients. However, in all genetic forms of the disease, the skin lesions are similar in appearance, evolution, and localization, suggesting a similar patho-mechanism. In line with the ectodermal nature of this condition, affected individuals may also present nail dystrophy and spared hair. Notably, 10% of patients with primary mitochondrial disorders may present with skin symptoms given the importance of the mitochondrial energy metabolism in skin homeostasis [72]. 

Ocular findings. Microphthalmia is only observed in LSDMCA1 and can affect one or both eyes. It is observed in 77% of LSDMCA1 with Xp22 rearrangements and 100% of cases with *HCCS* point mutations (Table 2). The phenotype can be very severe and progress to anophthalmia. Additional ocular findings include sclerocornea (unilateral or bilateral), corneal opacity, prolapsed iris, orbital cysts, cornea plana, hypoplasia of the optic nerve, aphakia, anterior eye chamber defects, coloboma, microcornea, hypopigmented and disorganized retinal pigmented epithelium, cataracts, choroidal thickening, chorioretinopathy, glaucoma, lens abnormalities, aniridia, pale optic disk, and altered visual-evoked potential. As shown in Table 2 and Appendix A, additional ocular signs can be seen also in LSDMCA2 and 3, although they are more common in LSDMCA1. 

Central nervous system (CNS) involvement. Regardless of the underlying genetic cause, the CNS is frequently implicated (in ~60% of LSDMCA1 cases, in the majority of LSDMCA2 patients, and in 1/3 LSDMCA3 cases) with a variety of clinical signs and symptoms. CNS anomalies include agenesis or hypoplasia of the corpus callosum, abnormal myelination, colpocephaly, seizures, hydrocephalus, ventriculomegaly, cysts, malformation of the septum pellucidum, and occult spinal dysraphism. Anencephaly has been described in a few cases (mainly aborted fetuses). Microcephaly can be observed as well as psychomotor developmental delay/intellectual deficits. Autistic behavior was described in a case with a 12.9-Mb Xp terminal deletion. Attention-deficit/hyperactivity disorder (ADHD) was diagnosed in a child with a point mutation in *COX7B* (see Appendix A).

Cardiac findings. Heart abnormalities are observed in all forms of LSDMCA. They have occurred in 36% of LSDMCA1, in two individuals with point mutations in *COX7B*, and in two cases with point mutations in *NDUFB11*. Clinical features include atrioventricular septal defects, patent ductus arteriosum, coarctation of the aorta, patent foramen ovale, ventricular tachycardia, atrioventricular block, histiocytoid cardiomyopathy, poor contraction of the left ventricle, and eosinophilic cell infiltration. The two patients with point mutations in *COX7B* displayed tetralogy of Fallot and ventricular hypertrophy, pulmonary hypertension, and atrial septal defect [3]. Finally, patients bearing point mutations in *NDUFB11* presented with histiocytoid cardiomyopathy and dilated cardiomyopathy and needed heart transplantation at the age of 6 months [4]. Notably, mutations in *NDUFB11* have also been described in patients affected only by histiocytoid cardiomyopathy without features of LSDMCA [73,74], in a male infant with lethal mitochondrial complex 1 deficiency [75] and in sideroblastic anemia [76,77].

Other clinical findings include diaphragmatic hernia, which may also result in respiratory distress, and short stature. Facial dysmorphisms, genitourinary defects (intersexual genitalia, hypoplastic genitalia, imperforate or displaced anus and polycystic ovary syndrome), and skeletal abnormalities are mainly seen in LSDMCA1 cases with Xp22 rearrangements, possibly due to the involvement of other genes responsible for X-linked disorders [8,78,79,80,81,82]. An exception to this observation is represented by a patient displaying a point mutation in *COX7B* presenting with an asymmetric face with limited eyelid closure, a small chin, left renal agenesis, and ureteral duplication of the right kidney [3]. Figure 1 depicts the typical linear skin lesions and some of the facial dysmorphism observed in LCDMCA.

Variable expressivity in LSDMCA. Extensive variability in the phenotypic expressivity, ranging from very mild or no phenotype to severe clinical manifestations, has been described between individuals and even within the same family. Several examples are available for LSDMCA1. Allanson and Richter described a female with typical skin and ocular clinical manifestations whose mother, bearing the same Xp-terminal deletion, was healthy, except for areas of depigmented skin on the shoulder and the leg, which were recognized only after examination with ultraviolet light [71]. More recently, Vergult et al. reported a familial case in which both the mother and the daughter presented a submicroscopic deletion of 185-220 kb on Xp22.2. Both patients presented microphthalmia that in the mother was severe and required eye enucleation. Instead, the skin lesions were only observed in the mother, who also suffered three spontaneous abortions of unknown sex within the first trimester [67]. Moreover, it should be noted that in LSDMCA1, the severity of the phenotype is not strictly related to the extent of the Xp-chromosome deletion that represents the underlying genetic cause in the majority of cases. Molecular characterization indeed demonstrated that patients with very large deletion of distal Xp, such as patient 2 from Lindsay et al., show a mild phenotype [1] while patients with point mutations in HCCS display the full phenotype [2,13] (Appendix A). A specific patient only displayed the typical linear skin defects, presented a 46,X,del (X) (pter-p22.2) karyotype, and was referred to the genetics clinic following the abortion of an anencephalic female fetus with the same karyotype [1]. Concerning LSDMCA2, only four cases have been reported to date. However, in the described familial case, a heterozygous nonsense mutation in the *COX7B* was identified in individual II.4 and her mother. The former presented the classical skin lesions, microcephaly, facial dysmorphisms, tetralogy of Fallot, clinodactyly of the fifth finger, intellectual disabilities, CNS malformation, poor vision, and ophthalmologic findings. Meanwhile, the mother only displayed the skin phenotype, a mild myopia, and had reported three pregnancies that ended with spontaneous abortions of unknown sex within the first trimester [3]. Lastly for LSDMCA3, only four cases have been reported and three are from a familial case. Subject 2 showed the typical linear skin lesions, ocular findings not including microphthalmia or sclerocornea, severe developmental delay, short stature, microcephaly, severe hypotonia, delayed dentition, brain malformations, and dilated cardiomyopathy that required heart transplantation. The mother, in whom the same mutation was identified, was completely asymptomatic. In the next pregnancy, ultrasound examination diagnosed a female fetus with a severe cardiological and neurological phenotype and intrauterine growth retardation that led to termination of pregnancy. Molecular studies identified the same frameshift mutation found in both the mother and subject 2 [4].

Concerning the possibility that the type of mutation could have an effect on the variability of the phenotype observed in LSDMCA, with the exception of those found in *HCCS*, all mutations identified in *COX7B* and *NDUFB11* are frameshift or premature STOP codon (see Table 1) [3,4] that most likely represent loss-of-function mutations. Similarly, the two missense mutations involving highly conserved amino acids identified in *HCCS* (Table 1) [2,19] display a loss-of-function effect given their inability to complement a *S. cerevisiae* strain deficient for the HCCS orthologous gene product [19]. 

In particular, for *COX7B*, the c.196delC mutation leads to the generation of a COOH-terminal truncated protein with the insertion of 46 novel amino acids. This mutant form is predicted to lack the domain that interacts with the two COX subunits, namely COX4 and COX6C [3,26]. The heterozygous splice mutation (c.41-2A>G) creates a novel splice acceptor site, leading to the production of out-of-frame transcripts and most likely to the formation of a truncated protein with 18 novel amino acids at the COOH-terminal region [3]. Finally, the heterozygous c.55C>T nonsense mutation may result either in a functional null allele as result of nonsense-mediated mRNA decay or in a protein lacking a large C-terminal portion (Δaa19–80)[3]. Notably, this mutation has been identified in the patient and her healthy mother [3], suggesting that the severity of the LSDMCA phenotype is not due to the consequence of different mutations identified in COX7B, and implying that other mechanisms are responsible for the high clinical variability observed in LSDMCA.

Similarly, the two point mutations identified in *NDUFB11,* i.e., 262C>T and c.402delG, are predicted to lead to the introduction of a premature stop codon and to a frameshift, respectively [4]. The latter mutation (p.Arg134Serfs∗3) was identified in a patient, a severely malformed fetus, and their healthy mother and was also identified in patients with histiocytoid cardiomyopathy without skin manifestations, further confirming the absence of a genotype/phenotype correlation [4,73]. Finally, heterozygous and hemizygous mutations in *NDUFB11* were also associated with other X-linked conditions [75,76,77,83]. Notably, all females carry a loss-of-function heterozygous variant, whereas males carry a hemizygous missense or in-frame deletion, indicating that residual activity of NDUFB11 is needed for males to be viable [83].

## 4. The Role of X-Chromosome and X-Chromosome Inactivation

All the genes so far involved in this rare genetic condition, i.e., *HCCS*, *COX7B,* and *NDUFB11,* are localized on the X chromosome. This segment of our genome has many peculiar features, including X-chromosome inactivation (XCI), also known as “lyonization”. This phenomenon consists of the transcriptional silencing of one of the two X chromosomes in female mammals to achieve dosage compensation between sexes [84]. This epigenetic process starts at the blastocyst stage in the early phases of embryo development. In normal conditions, the choice of which of the two X chromosomes is to be silenced is random but is then maintained in the progeny cells. Thus, in normal conditions, the ratio of the two-cell population (carrying the active and the inactive X, respectively) is around 50:50. As a consequence, normal female individuals are naturally mosaic and display organs with a mixed population of cells in which either the paternal or the maternal X has been inactivated. For this reason, women are less susceptible to pathogenic variants on the active X chromosome as the variant will not be expressed in all cells [85]. This explains why LSDMCA female patients can be observed while nullisomy for these genes in the hemizygous males is lethal as affected fetuses do not survive and are aborted. On the contrary, in diseased conditions, when one of the X chromosomes displays a mutation, the choice of which of the two chromosomes will be silenced is not random: the normal X chromosome is favored and the mutated X is preferentially inactivated. In this case, we have a divergence from the 50:50 ratio that is known as skewing of the XCI that can occur with variable degrees (Figure 2). The extensive intrafamilial and interfamilial phenotypic variability observed in this condition can be explained by X-inactivation. We hypothesize that in the affected heterozygous females, once the XCI process takes place in the early stages of embryo development, cells inactivating the normal X chromosome will die as a consequence of loss-of-function mutations in *HCCS*, *COX7B,* or *NUFB11*, on the transcriptionally active X chromosome [25,86]. We propose therefore that the clinical signs observed in LSDMCA would be the consequence of the different capabilities of the diverse tissues and organs to remove the “affected dying” cells by cell selection. According to our hypothesis, individuals characterized by a mild phenotype or even the total absence of clinical manifestations are the result of a completely skewed inactivation that obligates preferential inactivation of the mutated X chromosome, leaving the non-mutated, transcriptionally active X in blood cells and tissues (e.g., eyes and skin). Conversely, the most severe clinical manifestations can be found in patients in which the mutated X chromosome is active while the non-mutated one is transcriptionally silenced in tissues affected by the disease and/or at determined developmental stages (Figure 2).

## 5. LSDMCA as an Unconventional Mitochondrial Disorder 

Mitochondrial diseases are a clinically heterogeneous group of rare disorders resulting from an MRC dysfunction. *HCCS*, *COX7B,* and *NDUFB11* encode proteins necessary for the proper function of the MRC, thus defining LSDMCA as a mitochondrial disorder. 

Although clinically heterogeneous, the typical features of mitochondrial diseases include neuromuscular hypotonia, ataxia, encephalopathy, encephalomyopathy, and various myopathies [87,88,89]. Interestingly, as previously described, LSDMCA patients show diverse phenotypes, mainly characterized by developmental defects affecting the eyes and the skin and, in this regard, LSDMCA represents an unconventional mitochondrial disease. LSDMCA represents a remarkable example of a truly developmental phenotype associated with mitochondrial dysfunction. Although neurological disorders and cardiac defects are common features of mitochondrial disorders, the unique skin lesions mainly affecting the head and neck and the microphthalmia are difficult to assign to deficiency of mitochondrial enzymes. Interestingly, *HCCS* was the first human gene encoding for an MRC protein which causes microphthalmia when mutated. Moreover, the skin involvement observed in mitochondrial diseases is atypical and can include hirsutism and hypertrichosis, as described in Leigh syndrome (OMIM # 256000), and twisted hairs as reported in Bjornstad syndrome (OMIM # 262000) [90]. 

*HCCS*, *COX7B,* and *NDUB11* are all ubiquitously expressed since they are required for the OXPHOS pathway. However, the phenotypic manifestations observed in LSDMCA-affected females mainly affect the CNS. The remarkable involvement of the CNS is not surprising when considering the necessity for this high-energy-demand system to preserve adequate mitochondria number and function, a process which is termed mitostasis [91]. 

More recently, patients with *NDUFB11* mutations displaying more classical mitochondrial phenotypes have been described [75,83]. These patients show a combination of neurological symptoms, muscle hypotonia, myopathy, lactic acidosis, histiocytoid cardiomyopathy, and sideroblastic anemia. Interestingly, these patients are all males carrying a missense or one amino acid in-frame deletion variants [83]. This observation indicates that some residual activity of the NDUFB11 protein may explain the less severe phenotype and is needed for males to be viable. 

It is possible that differential tissue sensitivity to mitochondrial ATP depletion (high versus low energy demand) and/or overproduction of reactive oxygen species might elicit different molecular responses in the absence of *HCCS*, *COX7B,* or *NDUFB11* in selected tissues and may therefore induce the blockages to cell replication and/or increased cell death. In addition, although the main function of the mitochondrion is the production of energy in the form of ATP, it is well known to have a central role in the regulation of the intrinsic pathway of cell death, a key process required for proper development of the CNS [92,93]. Deregulation of these processes and X-chromosome inactivation could function together to select OXPHOS-proficient cells, thus attenuating or abolishing MRC defects in the surviving tissues and individuals. 

Nevertheless, additional studies are required to better understand the peculiarity of the LSDMCA phenotype that, to date, represents a unique example of a mitochondrial disease mainly characterized by an apoptosis-driven developmental phenotype. 

## Figures and Tables

**Figure 1 genes-12-00263-f001:**
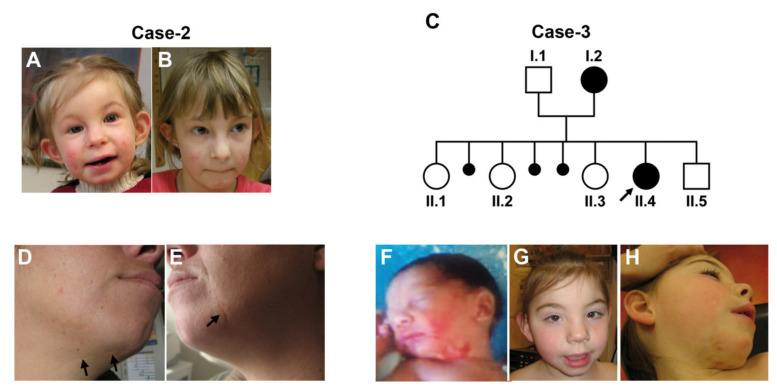
Linear skin lesions in LSDMCA2 patients. (**A**,**B**) Case-2. She had asymmetric face with limited eyelid closure, linear skin defects on face and neck, which became less obvious with age ((**A**): age 1^10/12^ years; (**B**): age 7 years). (**C**) Pedigree of Case-3. Parents are asymptomatic and unrelated. Three pregnancies ended with abortions. (**D**–**H**) Photographs of Case-3 (II.4) and her mother (I.2). Individual I.2, presented with linear skin defects, which healed with scarring (arrows) (**D**,**E**). Individual II.4 had facial dysmorphism with telecanthus, long upslanting palpebral fissures, short nose, mild retrognathia, and posteriorly rotated ears (**G**). Linear and patchy erythrodermia on cheeks and neck, which were more pronounced at birth (**F**) compared to the age of 5 years (**H**). Hollow circles and squares indicate asyntomatic females and males, respectively. Solid circles indicate affected females. Small solid circles indicate aborted fetuses. Figure from [3] used with permission.

**Figure 2 genes-12-00263-f002:**
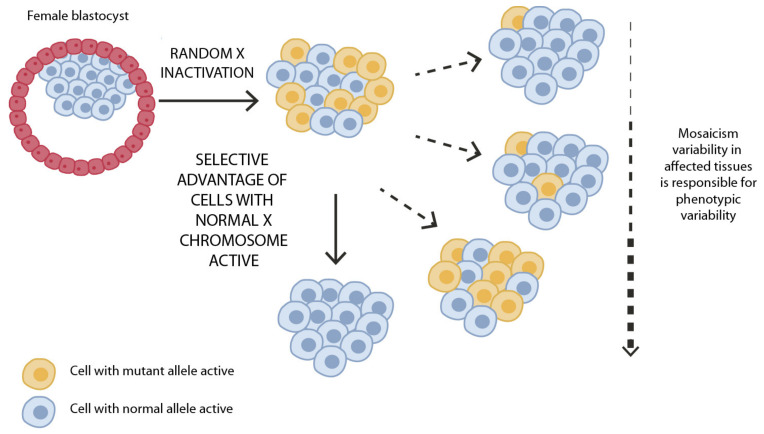
Schematic representation of XCI in female somatic cells. In normal conditions, the ratio of the two cell types (carrying the active and the inactive X chromosome) is approximately 50:50, but in females with X-linked dominant disorders, this ratio may be different, due to a potential disadvantage for cells expressing a mutant X-linked allele. Divergence from the 50:50 ratio, known as skewing of XCI, can be different in various tissues, in different developmental stages, and may vary among individuals, causing variability in the severity of the phenotype observed. For disorders such as LSDMCA, affected females usually have totally skewed patterns of XCI, in favor of an active wild-type X chromosome.

**Table 1 genes-12-00263-t001:** Summary of point mutations identified to date in LSDMCA.

Gene	Gene OMIM#	Nucleotide Change	Type of Mutation	Predicted Protein	Disease Symbol	Disease OMIM#	Ref
*HCCS*	300056	c.589C>T	Nonsense	p.R197* ^a^	*MLS/*	309801	[2,13,14]
		c.649C>T	Missense	p.R217C	*MIDAS*		
		c.475G>A	Missense	p.E159K	*MCOPS7*		
		c.[=/524_525delAG]	Frameshift	p.[=/E175Vfs*30]	*LSDMCA1*		
*COX7B*	300885	c.196delC	Frameshift	p.L66Cfs*48	*LSDMCA2*	300887	[3]
		c.41-2A>G	Frameshift	p.V14Gfs*19			
		c.55C>T	Nonsense	p.E19*			
*NDUFB11*	300403	c.262C>T	Nonsense	p.Arg88*	*LSDMCA3*	300952	[4]
		c.402delG	Frameshift	p.Arg134Serfs*3			

^a^ This mutation was identified in two different patients (see also Appendix A). Ref, references.

**Table 2 genes-12-00263-t002:** Summary of clinical findings found in LSDMCA. Extended details can be found in Appendix A.

LSDMCA/ Mutation		Skin Lesions	EYE			CNS Malformations	Intellectual Disabilities	Short Stature	Cardiac Anomalies	Ref
			Micro/Anophthalmia	Corneal Abnormalities	Other					
LSDMCA1/ Xp22 R	# cases	54/70	54/70	45/70	31/70	37/65	13/46	23/49	23/65	[1,2,5,13,15,33,34,35,36,37,38,39,40,41,42,43,44,45,46,47,48,49,50,51,52,53,54,55,56,57,58,59,60,61,62,63,64,65,66,67,68,69,70,71]
	%	77	77	64	44	57	28	47	35	
LSDMCA1/ *HCCS* P	# cases	2/5	5/5	5/5	3/5	3/5	3/5	1/5	2/5	[2,13]
	%	40	100	100	60	60	60	20	40	
LSDMCA2/ *COX7B* P	# cases	4/4	0/4	0/4	1/4	3/4	2/4	2/4	2/4	[3]
	%	100	0	0	25	75	50	50	50	
LSDMCA3/ *NDUFB11* P	# cases	2/3	0/4	0/4	2/3	1/3	1/3	1/2	2/3	[4]
	%	67	0	0	67	33	33	50	67	

For each group of patients, the top row indicates the total number of cases, the row below the percentage of patients displaying the indicated clinical sign. Abbreviations: Ref, references; R, rearrangements; P, point mutations.

## Data Availability

Data sharing not applicable.

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
