# Peer review of "Linear Skin Defects with Multiple Congenital Anomalies (LSDMCA): An Unconventional Mitochondrial Disorder"

_genes, 2021, doi:10.3390/genes12020263_

Round 1

Reviewer 1 Report

This review is well written and gives a nice overview of LSDMCA disease. In particular, the chapter "The clinical spectrum of LSDMCA", table 1, and supplementary table 1 describe in detail the symptoms of the disease and offer an in-depth understating of the phenotypic and clinical aspects of LSDMCA. Moreover, the chapter "The role of X-chromosome and X-chromosome inactivation" gives an exhaustive explanation of the variability of symptoms that can be found in LSDMCA patients.

Minor suggestions:

Row 39, 47, 124, and 129: I suggest the authors to remove "(see below)".

Row 111-113 "Although HCCS is a ubiquitous protein, increased expression levels have been detected in the heart, skeletal muscles, and the Central Nervous System (CNS), including the eye, suggesting a tissue/cell type-specific requirement for this protein": no reference was included. Moreover, by checking the Human Protein Atlas database, HCCS protein is expressed at low level in CNS and muscles, and it is not expressed in the eye (https://www.proteinatlas.org/ENSG00000004961-HCCS/tissue). The authors should add a proper reference or remove that sentence. 

Row 113-115 "Hccs migrates from mitochondria to the cytosol under apoptotic stimuli resulting in suppression of the X-linked inhibitor of apoptosis (XIAP) protein and activation of cell death": authors should add more informations. The pathway they described is not well known. However (and as an example), holocytochrome c is known to form a complex with Apaf-1, thus forming the apoptosome complex, which recruit and activate procaspase-9 (https://doi.org/10.1038/sj.cdd.4400782). While, SMAC is released in the cytosol together with cytochrome c and neutralizes the X-linked inhibitor of apoptosis protein (XIAP) (https://doi.org/10.1038/cdd.2017.179). 

Row 117: they wrote medaka fish instead of medakafish, as elsewhere in the text.

Row 154-156 "Only 14 proteins
155 represent the core subunits and are essential for energy transduction, whereas the role of the 30 supernumerary subunits are still poorly understood": I suggest the author to add the paper "Mammalian Mitochondrial Complex I Structure and Disease-Causing Mutations" (DOI: 10.1016/j.tcb.2018.06.006), published in 2018. It describes very well the complex I subunits and the mutations associated with human diseases.  

Line 341-342 "“Canonical” mitochondrial diseases are thus usually characterized by postnatal organ failure": I suggest the authors to eliminate the sentence. Mitochondrial diseases are very heterogeneous. 

Chapter "LSDMCA as an unconventional mitochondrial disorder": The authors should discuss the skin defects in other mitochondrial disorders. For example, "Mitochondrial dysfunction: a neglected component of skin diseases" (DOI: 10.1111/exd.12484) affirm that "Ten per cent of patients with primary mitochondrial disorders present skin manifestations that can be categorized into hair abnormalities, rashes, pigmentation abnormalities and acrocyanosis"

Line 354-355 "However, the phenotypic manifestations observed in LSDMCA-affected females mainly affect the CNS": the authors should better elaborate this sentence. Phenotypic manifestations mainly affect the CNS because brain metabolism mainly depend on OXPHOS and ATP production (DOI: 10.1016/j.neuron.2017.09.055). This is why a lot of neuropsychiatric, neurodegenerative, and neurodevelopmental disorders are associated with mitochondrial dysfunctions. 

Line 366-369 "In addition, although the main function of the mitochondrion is the production of energy in the form of ATP, it is well known to have a central role in the regulation of the intrinsic pathway of cell death, a key process required for proper development of the CNS": a reference should be provided. 

Reviewer 2 Report

Indrieri and colleagues summarized three subtypes of LSDMCA and discussed the genes and phenotypes associated with them. Comments and suggestions are below.

Major:

  1. The structure of this review article can be condensed and more meaningful, especially when describing the clinic presentations.
  2. Though mutations occurred on the same gene, the consequence of different mutations may vary dramatically. Are they loss-of-function mutations or partial loss-of-function mutations? These may directly affect the severity of phenotypes.
  3. Since HCCS, COX7B and NDUFB11 are all mitochondrial related genes, it would be nice to compare the mitochondrial function studied with each mutation in a table. For example, reduced OXPHOS? Impaired 12S rRNA and mitochondrial mRNAs expression?
  4. Figure 1, why not start with Case 1? In panel C, Case 3-I.2 (mother of Case-3-II.4) was identified as a patient while in the figure legend, it states “Parents are healthy and unrelated”.

Minor:

  1. Column names are missing in Tables 1 and 2.
  2. Line 234, “COX7B”should be italic.

Round 2

Reviewer 2 Report

I don't have any other suggestions to the authors. This manuscript is suitable for publication.